# Identification of microbial antigens in liver tissues involved in the pathogenesis of primary biliary cholangitis using 16S rRNA metagenome analysis

Tomohiro Katsumi[1]*, Hidenori Sato[2], Ryoko Murakami[2], Takumi Hanatani[1], Fumi Uchiyama[1], Fumiya Suzuki[1], Keita Maki[1], Kyoko Hoshikawa[1], Hiroaki Haga[1], Takafumi Saito[1], Yoshiyuki Ueno[1]

1 Department of Gastroenterology, Faculty of Medicine, Yamagata University, Yamagata, Japan, 2 Genomic Information Analysis Unit, Department of Genomic Cohort Research, Faculty of Medicine, Yamagata University, Yamagata, Japan

* t-katsumi@med.id.yamagata-u.ac.jp

**Data Availability Statement:** "All relevant data are within the paper and its Supporting Information files."

## Abstract

### Background

Multiple factors are involved in the pathogenesis of primary biliary cholangitis (PBC), a chronic cholestatic liver disease, characterized by intrahepatic cholangiopathy. In particular, studies have suggested that environmental factors such as the presence of granulomas in the portal vein region are important for the development of PBC. This study aimed to comprehensively analyze and identify foreign-derived antigens in PBC liver tissue to confirm their involvement in PBC pathogenesis.

### Methods

Portal areas and hepatocyte regions were selectively dissected from formalin-fixed paraffin-embedded PBC liver tissue samples using the microlaser method, followed by total DNA extraction. We then validated whether the bacterial strains identified through 16S rRNA metagenomic analysis were detected in PBC liver tissues.

### Results

The most frequently detected bacterial genera in the PBC liver tissue samples were *Sphingomonas panacis*, *Providencia*, and *Cutibacterium*. These bacterial genera were also detected in the other PBC samples. Validation for the detection of *S. panacis*, the most abundant genus, revealed polymerase chain reaction bands extracted from the portal areas of all samples. They were also more highly expressed than bands detected in the hepatocyte region.

### Conclusion

*S. panacis* antigen was specifically detected in the portal areas of PBC liver tissues. The introduction of foreign-derived antigens into the liver as an environmental factor could be a possible mechanism for the development of PBC.

**Funding:** Health and Labour Sciences Research Grants for Research on Measures for Intractable Diseases (from the Ministry of Health, Labour and Welfare of Japan), a Grant-in-Aid for Scientific Research C (23K07409) from JSPS.

**Competing interests:** The authors have declared that no competing interests exist.

## Introduction

Primary biliary cholangitis (PBC) is a chronic cholestatic liver disease characterized by pathological intrahepatic small bile duct damage, the pathogenesis of which remains unknown [1–5]. Multiple factors are believed to be involved in the pathogenesis of PBC, with autoimmune mechanisms being considered the main triggering factors, considering that PBC is often associated with chronic thyroiditis and Sjogren's syndrome [6–10].

In recent years, extensive genome-wide association studies have been conducted to identify PBC disease-specific genes [11–13]. Other studies identified a locus on the X chromosome that may be involved in the female predominance of PBC, suggesting a genetic predisposition to PBC [14]. Alternatively, a recent case–control study on Japanese patients with PBC and sex- and age-matched controls showed that environmental unsanitation and chronic exposure to chemicals (e.g., smoking and hair dye) during childhood may be risk factors for the development of PBC [15]. Therefore, these environmental factors may also play a major role in the development of PBC [2, 16].

Several types of PBC mouse models have been established, including models in which genetic factors are involved (e.g., NOD.c3c4 mice), models in which autoimmune mechanisms may be involved (e.g., dnTGF-βRII mice), and PBC-like pathologies caused by 2OA, 6BH immunization or *Escherichia coli* (*E.coli*) infection 2OA, and 6BH immunization [17]. In line with this, mice with a PBC-like pathology caused by *E. coli* infection suggest the involvement of environmental factors [18]. Granulomas found within the portal vein region of PBC liver tissues have been identified as key environmental factors [19, 20]. Specifically, studies have detected the presence of *Propionibacterium acnes* in PBC granulomas [21], identifying urinary tract infections as a risk factor for the development of the disease. Of particular interest is the finding that the corresponding antigen of anti-mitochondrial antibodies (AMA), the pyruvate dehydrogenase E2 component (PDC-E2), is conserved across species in these bacteria [22, 23]. This molecular homology strongly supports the role of environmental factors in the development of disease.

Although these studies have suggested that environmental factors are also important in the pathogenesis of PBC, the pathogenesis-associated exogenous antigens present in PBC organisms have not been continuously and universally identified. As such, identifying these exogenous antigens as environmental factors can help elucidate the pathogenesis of PBC, an intractable liver disease, and uncover future therapies.

The current study aimed to comprehensively analyze the exogenous antigens in PBC liver tissues and serum via 16S rRNA metagenome analysis and confirm their involvement in the pathogenesis of PBC.

## Materials and methods

### PBC sample selection

PBC cases were selected using the following diagnostic criteria: itchy skin, which is a typical clinical manifestation of PBC; elevated serum biliary enzymes and high IgM levels on blood biochemical tests; and positivity for AMA. In addition, cases of histopathologically confirmed PBC with interlobular bile duct lesions and granulomas were included in this study. Cases with histologically concomitant autoimmune hepatitis, severe fibrosis, or noncompensated cirrhosis were excluded. Cases with liver carcinogenesis in the background of PBC were also excluded. The control group comprised healthy volunteers. These PBC cases were diagnosed at Yamagata University Hospital, and liver tissue was used for the study only when permission to use the specimens was obtained from the patients.

## Selective laser microdissection

Laser microdissection (LMD), a method used for sectioning extremely small portions of a histopathological specimen, is an ideal technique for molecular analysis involving minute specimens. Formalin-fixed and paraffin-embedded (FFPE) sections of PBC liver tissues collected via liver biopsy were prepared on glass slides with foil (PEN-Membrane Slides, Leica) and stained with hepatoxylin eosin (HE).

From the FFPE liver tissue samples, the portal vein and hepatocyte regions were selectively excised using the LMD method (Leica LMD 6500, Leica MICROSYSTEMS, UV laser cutting) and collected in individual 0.5-mL polymerase chain reaction (PCR) tubes (PCR-05-C, Axygen). The hepatocyte area was selected and removed from the area without inflammatory cell infiltration. The laser was adjusted to a wavelength of 349 nm to minimize sample damage. The area of the excised portal vein and hepatocyte regions per sample was measured.

## DNA extraction from liver tissues and sera

DNA was extracted from the LMD-separated portal vein area and hepatocytes using the Mag-MAX FFPE DNA/RNA Ultra Kit (Thermo Fisher Scientific), followed by Quality check and concentration measurements (Qubit assays, Life Technologies). DNA was also extracted from the PBC sera for validation experiments (QIAamp MinElute Virus Spin Kit, QIAGEN). Concentration measurements and quality check were performed using the same procedure.

## 16S rRNA sequencing

Multiplex polymerase chain reactions were performed on the extracted DNA samples using two primer pools corresponding to the V2, V4, and V8 regions (amplicon sizes of 250, 288, and 295 bp, respectively) or the V3, V6–7, and V9 regions (amplicon sizes of 215, 260, and 209 bp, respectively) of the 16S rRNA gene (16S metagenomics kit, Thermo Fisher Scientific). Each hypervariable regions (V1-V9 regions) of 16s RNA demonstrate considerable sequence diversity among different bacteria and used for accurate identification. A sequence library was constructed using the IonAmpliSeq library kit (Thermo Fisher Scientific) according to the manufacturer's protocol. To ensure the quality of each library, digital electrophoresis was performed using a D1000 Screen Tape on a 2200 Tape Station (Agilent Technologies). Amplified 30–140 pM libraries were subjected to emulsion PCR, and purified libraries were loaded onto an Ion P1 chip v3, after which ion semiconductor sequencing was performed using an Ion Proton sequencer (Thermo Fisher Scientific). To obtain taxonomy assignment, >101 bp read lengths in sequence files were run through the assembly programs constructed using Bowtie2 version 2.3.5, along with consecutive mapping on the Greengenes database (published May 2011, including 16Sr RNA gene sequence information for 406,998 strains). The read depths against the 16S rRNA gene regions were calculated using bedtools software version 2.29.0 and the In House R script.

## Validation study

Validation PCR study was performed to detect the bacterial genera identified from the results in the 16S rRNA metagenome analysis results. Primers were designed refer to *Sphingomonas paucimobilis* and *S. panacis* reference sequences (NCBI accession AB055142, NZ_CP014168 respectively). The clustalw version 2.1 was used for alignment.

PCR (Agilent 4200 TapeStation, Agilent Technologies) was used to verify whether the bacterial genera identified via 16S rRNA metagenome analysis could be detected in PBC sera and

liver tissues using other cohort samples. In particular, we compared whether PCR detection levels differed between the portal vein and hepatocyte regions.

## Ethics statement

This study was approved by the research ethics committee of Yamagata University (Approval number: 2021-230) and written informed consent was obtained from all patients and all control subjects for serum and liver tissue prior to participation. The study was conducted in accordance with the Declaration of Helsinki.

All accessed data were fully anonymized for research purpose (date 22-Oct-2021).

## Statistics and data analysis

We used descriptive statistics to summarize the characteristics of PBC samples and Student's t-test was used to assess the statistical significance of differences.

## Results

### Clinical characteristics of the patients

The characteristics of the patients with PBC (n=24) included in this study are summarized in Table 1. As seen in the table, female predominance was apparent, with blood biochemical tests showing elevated transaminase, biliary enzymes, and high IgM levels. All cases tested negative for antinuclear antibodies. The liver tissues showed mild fibrosis.

### DNA library in PBC sera

Fig 1A shows that the quality check of the DNA library in *E. coli* (as a control) and PBC sera detected an amplification read peak at 200–300 bp, which was sufficient for 16S rRNA metagenome analysis. Fig 1B shows the read peak histogram in PBC samples. The final mean DNA concentration was 12 ng/μl for PBC and 15 ng/μl for *E. coli*.

**Table 1. Baseline characteristics of the patients with primary biliary cholangitis.**

| Characteristics | n=24 |
|---|---|
| Age, years | 54 (40-65) |
| Gender, female | 63% |
| **Biochemical examination** | |
| T.Bil (mg/dL) | 0.8 (0.5-1.1) |
| Alb (g/dL) | 4.1 (3.7-4.3) |
| AST (U/L) | 41 (24-75) |
| ALT (U/L) | 43 (15-70) |
| γ-GTP (U/L) | 270 (91-1201) |
| Plt (x$10^4$/μL) | 248 (198-278) |
| PT-INR | 0.96 (0.84-1.01) |
| AMA-M2 (index) | 227 (115-534) |
| IgM (mg/dL) | 360 (221-508) |

Parameters are presented as the mean (range). T.Bil, total bilirubin; Alb, albumin; ALT, alanine aminotransferase; AST, aspartate aminotransferase; γ-GTP, γ-glutamyl transpeptidase; Plt, platelet count; PT-INR, prothrombin time-international normalized ratio; AMA-M2, anti-mitochondrial M2 antibody; IgM, immunoglobulin M

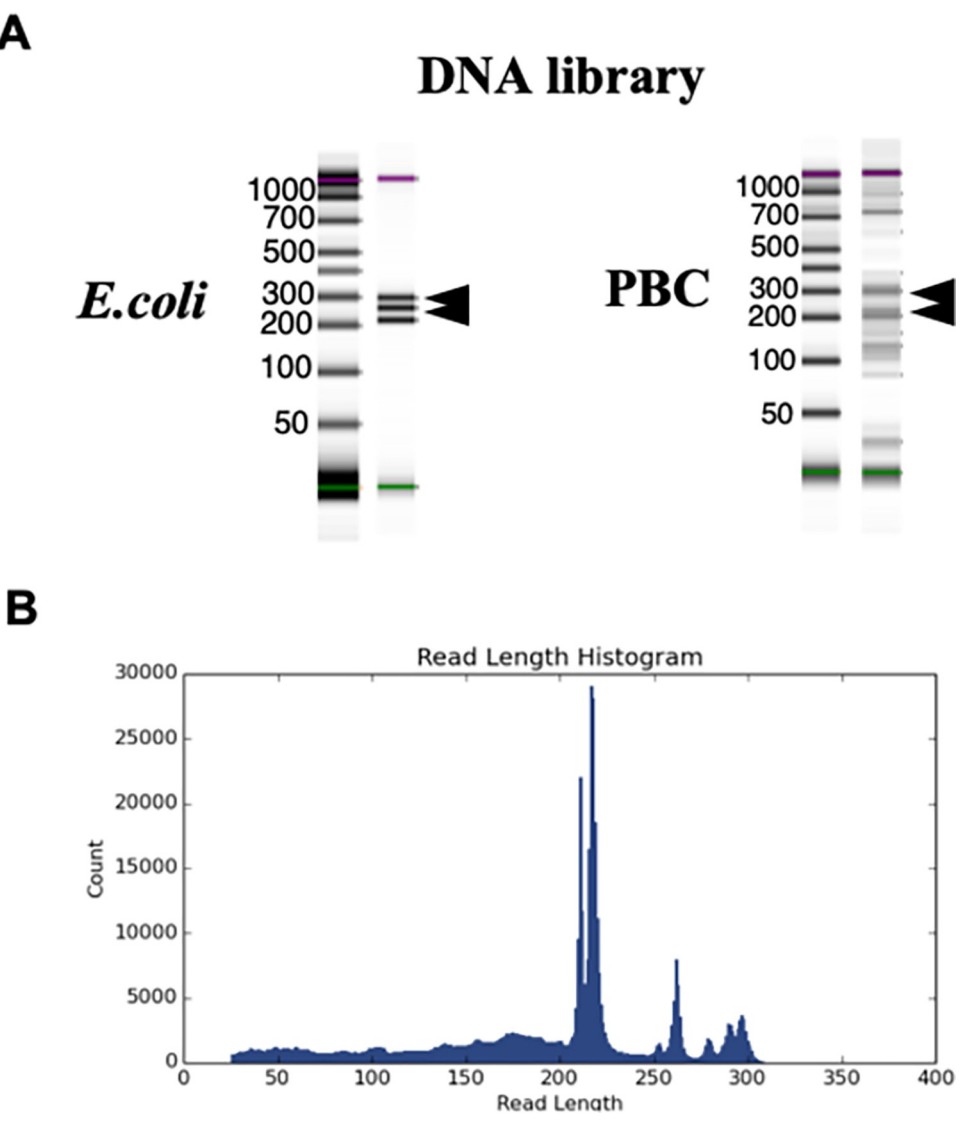

**Fig 1. DNA library in control and PBC sera.** (A) The quality check of DNA library in E.coli (as a control) and PBC sera detected an amplification read peak at 200–300 bp. (B) The Reads peak histogram in PBC samples. The results of the library check ensured the quantity and quality of DNA required for metagenomic analysis.

## PBC portal area and hepatocyte area after LMD

A total of 10 PBC liver biopsy samples were sectioned into portal and hepatocellular regions using LMD. Fig 2A shows the PBC liver histology before and after LMD of the portal area, whereas Fig 2B shows the PBC liver histology before and after LMD of the hepatocyte area.

Table 2 presents the total areas of the portal and hepatocellular regions. In all samples, the hepatocyte area was dissected more than the portal vein area. This was the volume of the samples collected, reflecting the proportion of the area in the liver tissue.

## 16S rRNA metagenome analysis in sera

The following is a list of the most common bacterial genera detected by metagenomic analysis of PBC sera (Table 3). In the PBC group, the most frequently detected reads were *Providencia* (mean 2578 reads), *Sphingomonas* (mean 2096 reads), *Cupriavidus* (mean 372 reads), and

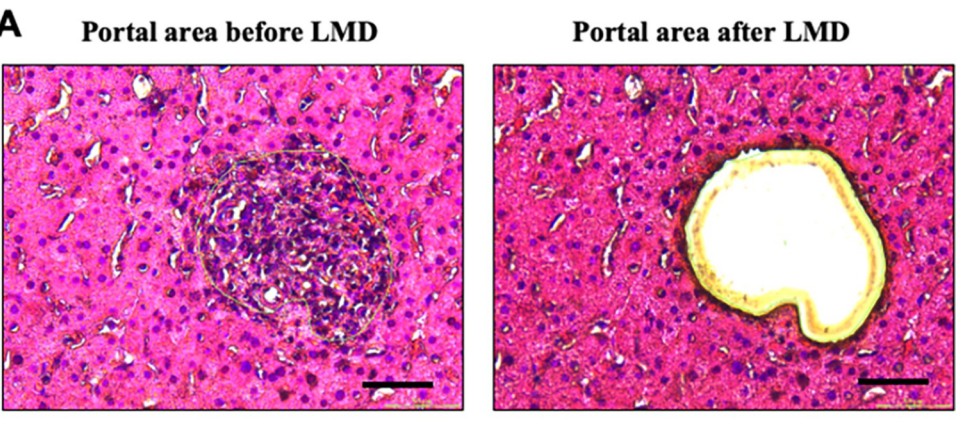

**Fig 2. Laser capture microdissection (LMD) of paraffin-embedded liver tissues from PBC portal area and hepatocyte area.** (A) The PBC liver histology before and after selective LMD of the portal area. (B) The PBC liver histology before and after LMD of the hepatocyte area. Each area was selectively and reliably dissected. Scale bare = 50 μm.

*Lactobacillus* (mean 359 reads). The reads detected in the healthy control group were *Providencia* (mean 28 reads), *Sphingomonas* (mean 9.5 reads), *Cupriavidus* (mean 2.5 reads), and *Lactobacillus* (mean 160 reads), all of which were significantly higher in the PBC sera. Each of these three alpha-proteobacteria was found to contain sequences of the 16S region in two or more locations, indicating their presence in PBC sera.

### Validation study in serum samples

*Sphingomonas paucimobilis*, the reference species of *Sphingomonas*, was used as a control for comparison in the validation PCR study. The gene sequences that could detect both *S. panacis* and *S. paucimobilis* reference species detected by 16S rRNA metagenome analysis were used as primers (S1 Table), whereas PCR was used to validate the results. Fig 3 shows that *S. panacis* was detected in the PBC sera of 2 out of the 11 cases (detection rate was 18%).

### 16S rRNA metagenome analysis in PBC liver tissues

16S rRNA metagenome analysis of 4 PBC liver tissue samples showed that more reads and bacteria were detected in both liver tissues than in the serum samples in Table 4. The most

**Table 2. The total areas of the dissected portal and hepatocyte regions after LMD.**

| sample | LMD region | LMD area (μm$^2$) |
|---|---|---|
| PBC1 | Portal | 30,535 |
| | Hepato | 186,429 |
| PBC2 | Portal | 27,558 |
| | Hepato | 280,424 |
| PBC3 | Portal | 50,156 |
| | Hepato | 255,432 |
| PBC4 | Portal | 140,554 |
| | Hepato | 169,123 |
| PBC5 | Portal | 91,137 |
| | Hepato | 212,061 |
| PBC6 | Portal | 15,526 |
| | Hepato | 58.926 |
| PBC7 | Portal | 16,410 |
| | Hepato | 173,873 |
| PBC8 | Portal | 38,409 |
| | Hepato | 150,780 |
| PBC9 | Portal | 16,339 |
| | Hepato | 126,735 |
| PBC10 | Portal | 24,666 |
| | Hepato | 148,715 |
| Total (mean) | Portal | 45,129 |
| | Hepato | 176,250 |

frequently detected bacterial genera were *S. panacis* (mean: 336,508 reads), *Providencia* (mean: 254,173 reads), and *Cutibacterium* (mean: 245,375 reads). These results were similar to those of the phylogenetic analysis of PBC sera, with the three most frequently detected bacteria being the same in all four tissue samples.

## Validation study in liver tissues

To validate the conventional PCR findings in PBC liver tissues, six samples were used to compare the detection sensitivity of the hepatocellular and portal regions. Fig 4A shows a comparison of the concentration of the PCR-amplified product (280 bp) of DNA extracted from the portal and hepatocellular regions. The PCR band extracted from the portal region was detected in all the samples. The concentration of *S. panacis* band detected in the portal region was higher than that of the band detected in the hepatocyte region in all samples. Furthermore,

**Table 3. The list of the most common bacterial genera detected by metagenomic analysis of PBC and healthy control sera (mean read counts).**

| Genus | Healthy Control (n=3) | PBC (n=24) | *p*-value |
|---|---|---|---|
| *Providencia* | 28 | 2578.1 | <0.001 |
| *Sphingomonas* | 9.5 | 2096 | <0.001 |
| *Cupriavidus* | 2.5 | 372.2 | <0.001 |
| *Lactobacillus* | 160.5 | 359.3 | <0.001 |
| *Aquabacterium* | 2.5 | 213.2 | <0.001 |
| *Spirosoma* | 6.5 | 200.1 | <0.001 |

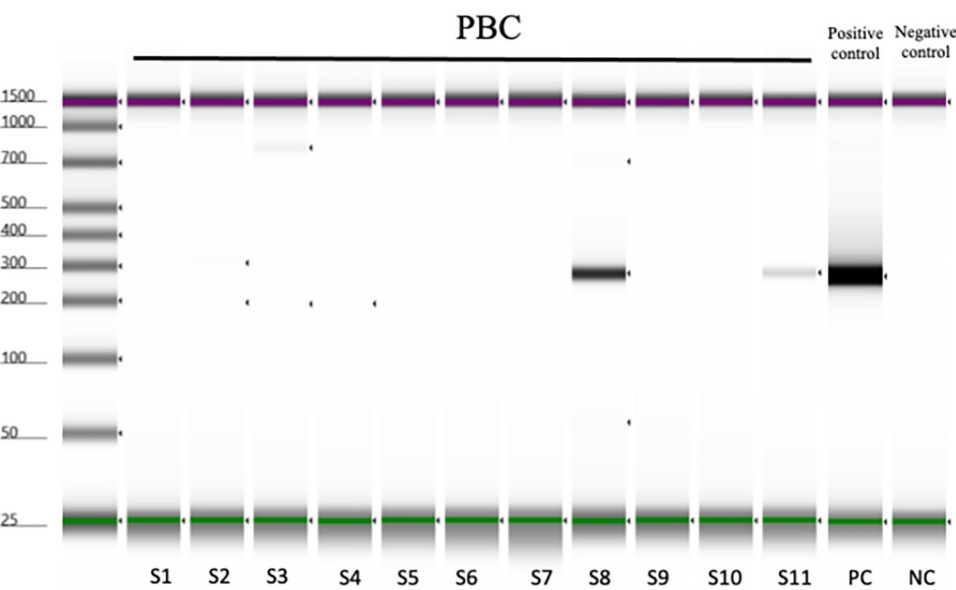

**Fig 3. Validation study in serum samples.** *Sphingomonas paucimobilis*, the reference species of *Sphingomonas*, was used as a control for comparison in the validation PCR study. The gene sequences that could detect both *S. panacis* and *S. paucimobilis* reference species. *S. panacis* was detected in the 2 PBC sera samples (sample ID: S8 and S11) and the detection rate was 18%.

when the concentration of the PCR-detected bands was standardized between the hepatocellular and portal areas of each LMD section, the PCR product extracted from the portal area was significantly more highly expressed than that from the hepatocellular area (Fig 4B).

## Discussion

PBC, drug-induced sarcoidosis, and viral hepatitis have been shown to promote the formation of epithelial granulomas in the liver [24–27]. Granulomas, which are composed of macrophage-derived epithelial cells and lymphocytes, exhibit the characteristics of a type IV allergy given that they are predisposed to bacterial infection and foreign body invasion. Therefore, we examined whether the formation of granulomas in PBC is triggered by foreign antigen invasion (i.e., the involvement of environmental factors in the development of PBC).

The current study evaluated PBC cases with typical histopathology and performed LMD selectively in the portal vein and hepatocyte areas of the liver tissue. The portal vein region contained granulomas, CNSDC (damaged bile ducts), and inflammation-inducing cells (mainly lymphocytes). After performing 16S rRNA metagenomic analysis to detect foreign-derived antigens, we found that *S. panacis* was the most frequently detected bacterial genus in the portal vein region. The highest number of reads was detected in all PBC liver tissue samples examined in this study. Furthermore, 16S rRNA metagenomic analysis of serum from PBC and controls showed that *S. panacis* was more frequently detected in the PBC group, although the number of detected reads was lower than that in liver tissue. Finally, *S. panacis* was detected specifically in the portal vein region of the liver tissues via PCR in both the portal vein and hepatocyte regions. Furthermore, even after standardization of the LMD cut-out area, the expression of *S. panacis* in the portal vein region was clearly different.

PBC is a multifactorial disease, indicating that several factors are involved in its pathogenesis, including autoimmune mechanisms, genetic predisposition, and environmental factors [2]. Among these factors, urinary tract infection has long been reported as a risk factor for the

**Table 4. The detected bacterial genera by 16S rRNA metagenome analysis in PBC liver tissues (mean read counts).**

|  | PBC 1 | PBC 2 | PBC 3 | PBC 4 |
|---|---|---|---|---|
| *Sphingomonas panacis* | 537,934 | 346,982 | 258,339 | 202,780 |
| *Providencia* | 324,427 | 347,571 | 183,687 | 161,007 |
| *Cutibacterium* | 233,392 | 388,344 | 126,194 | 233,570 |
| *Homo* | 36,807 | 43,660 | 71,712 | 14,843 |
| *Bradyrhizobium* | 14,768 | 11,841 | 12,684 | 13,179 |
| *Staphylococcus* | 14,117 | 32,936 | 20,907 | 13,426 |
| *Acinetobacter* | 6,517 | 4,401 | 914 | 841 |
| *Scardovia* | 6,092 | 13,545 | 5,553 | 1,471 |
| *Corynebacterium* | 5,370 | 13,563 | 31,572 | 4,394 |
| *Bacillus* | 3,289 | 4,331 | 1,836 | 1,808 |
| *Methylorubrum* | 3,213 | 3,211 | 1,224 | 1,308 |
| *Streptococcus* | 2,290 | 3,039 | 312 | 15 |
| *Burkholderia* | 1,963 | 2,579 | 991 | 1,335 |
| *Lawsonella* | 1,814 | 6,280 | 20,324 | 8,989 |
| *Bartonella* | 1,784 | 1,605 | 772 | 676 |
| *Brevundimonas* | 1,682 | 30 | 10 | 26 |
| *Gluconobacter* | 1,518 | 2,164 | 2,309 | 334 |
| *Candidatus_hamiltonella* | 1,216 | 108 | 2 | 3 |
| *Ralstonia* | 1,064 | 381 | 15 | 1,237 |
| *Planococcus* | 987 | 2,623 | 755 | 1,770 |
| *Thalassospira* | 979 | 1140 | 774 | 528 |
| *Devosia* | 881 | 714 | 520 | 51 |
| *Clostridium* | 700 | 555 | 463 | 279 |
| *Chloracidobacterium* | 647 | 480 | 262 | 188 |
| *Paenibacillus* | 586 | 1,026 | 4 | 4 |
| *Segniliparus* | 580 | 2,517 | 14 | 1 |
| *Stenotrophomonas* | 579 | 435 | 288 | 194 |

development of PBC. *E.coli* has been the predominant causative pathogen in most cases of urinary tract infections. Moreover, ample evidence has shown that *E.coli* infection is an important factor in disrupting immunological tolerance of mitochondria through molecular mimicry between human and *E.coli* E2 subunits of the 2-oxoacid dehydrogenase complex, the major autoantigen for AMA, leading to the production of disease-specific autoantibodies for AMA and PBC [28]. Studies have suggested that bacterial infections, such as *E.coli*, which retain PDC-E2 molecular homology, may induce immune abnormalities by introducing foreign-derived antigens into the body. *S. panacis*, which was highly expressed, is a member of the genus *Sphingomonas*, which is the same genus as *Novosphingomonas* previously reported to be detected in PBC sera. These results strongly indicate that molecular homology in the corresponding antigens and the mechanism of immune tolerance breakdown after infection are involved in the PBC pathogenesis. *S. panacis*, a bacterium normally found in soil and other sources, may be pathogenic [29]. In particular, cohort studies examining PBC environmental factors have reported differences in early childhood hygiene as a risk factor for developing PBC, which may be associated with the environment to which *S. panacis* is likely to be exposed. Sustained bacterial exposure may promote persistent in vivo entry of exogenous antigens, which may cause excessive antibody production and induction of inflammation by AMA-responsive antigens.

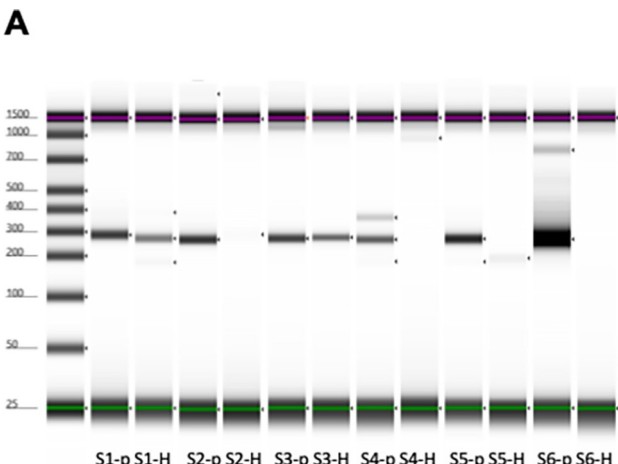

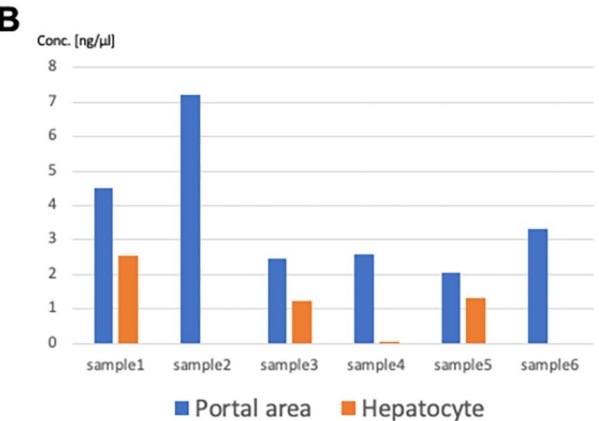

**Fig 4. Validation study in liver tissues.** (A) A comparison of the concentration of the PCR-amplified product (280 bp) of DNA extracted from the portal and hepatocellular regions. The PCR band extracted from the portal region was detected in all the samples. The concentration of S. panacis band detected in the portal region was higher than that of the band detected in the hepatocyte region in all samples. (B) The concentration of the PCR-detected bands was standardized between the hepatocellular and portal areas of each LMD section, the PCR product extracted from the portal area was significantly more highly expressed than that from the hepatocellular area.

*S. panacis*, newly detected in this study, is consistent with previous reports at the genus level and appears to be an important bacterium involved in environmental factors of hygiene and infection.

In addition, selective LMD of the portal and hepatocellular regions of PBC liver tissues demonstrated that *S. panacis* was rarely expressed in the hepatocellular region and expression was restricted to the portal region. This could be a possible mechanism by which foreign-derived antigens enter the liver via the important hepatic inflow pathways of the hepatic artery, portal vein, and lymphatic vessels and trigger the development of PBC. Granulomas are type IV allergic reactions that are typically unresponsive to the original antigen in vivo. Therefore, granuloma formation can be attributed to the influx of antigens from outside the body via a pathway wherein the initial reaction, such as an infection, culminates in an allergic reaction. Given that the portal region is home to various cells that engage in dynamic reactions, large amounts or chronic persistent exposure to foreign antigens could likely cause abnormal reactions.

A limitation of this study was the small number of cases in the validation experiment. Further validation of the number of cases is needed.

The current study showed that PBC may be triggered by bacteria. Elucidating the pathophysiology and pathogenesis of PBC, an intractable liver disease, could facilitate the development of effective treatments that promote better outcomes in the future.

## Supporting information

**S1 Table. The primers to detect both *S. panacis* and *S. paucimobilis* reference species.**
(DOCX)

**S2 Table. LMD raw data to show the area of portal site and hepatocyte after LMD.**
(XLSX)

**S3 Table. 16S rRNA metagenome raw data.** Whole genome sequencing data are contained.
(TXT)

**S4 Table. PBCserum 16S rRNA metagenome raw data.**
(TXT)

**S5 Table. PBC FFPE 16S rRNA metagenome raw data.**
(TXT)

**S1 Raw images. PCR gel raw images.**
(PDF)

## Author Contributions

**Formal analysis:** Hidenori Sato, Ryoko Murakami.

**Investigation:** Tomohiro Katsumi, Ryoko Murakami, Takumi Hanatani, Fumi Uchiyama, Fumiya Suzuki, Kyoko Hoshikawa, Hiroaki Haga, Takafumi Saito.

**Software:** Ryoko Murakami.

**Supervision:** Takafumi Saito, Yoshiyuki Ueno.

**Validation:** Keita Maki.

**Writing – original draft:** Tomohiro Katsumi.

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
