## [Decision Letter · Decision Letter 0]

9 Apr 2024

PONE-D-24-05687Identification of microbial antigens in liver tissues involved in the pathogenesis of primary biliary cholangitis using 16S rRNA metagenome analysisPLOS ONE

Dear Dr. Katsumi,

Thank you for submitting your manuscript to PLOS ONE. After careful consideration, we feel that it has merit but does not fully meet PLOS ONE’s publication criteria as it currently stands. Therefore, we invite you to submit a revised version of the manuscript that addresses the points raised during the review process.

We look forward to receiving your revised manuscript.

Kind regards,

Hin Fung Tsang

Academic Editor

PLOS ONE

Journal Requirements:

2. Thank you for submitting the above manuscript to PLOS ONE. During our internal evaluation of the manuscript, we found significant text overlap between your submission and previous work in the [introduction, conclusion, etc.].

Please revise the manuscript to rephrase the duplicated text, cite your sources, and provide details as to how the current manuscript advances on previous work. Please note that further consideration is dependent on the submission of a manuscript that addresses these concerns about the overlap in text with published work.

[If the overlap is with the authors’ own works: Moreover, upon submission, authors must confirm that the manuscript, or any related manuscript, is not currently under consideration or accepted elsewhere. If related work has been submitted to PLOS ONE or elsewhere, authors must include a copy with the submitted article. Reviewers will be asked to comment on the overlap between related submissions (http://journals.plos.org/plosone/s/submission-guidelines#loc-related-manuscripts).]

We will carefully review your manuscript upon resubmission and further consideration of the manuscript is dependent on the text overlap being addressed in full. Please ensure that your revision is thorough as failure to address the concerns to our satisfaction may result in your submission not being considered further.

"Health and Labour Sciences Research Grants for Research on Measures for Intractable Diseases (from the Ministry of Health, Labour and Welfare of Japan), a Grant-in-Aid for Scientific Research C (23K07409) from JSPS"

"This study was supported in part by Health and Labour Sciences Research Grants for Research on Measures for Intractable Diseases (from the Ministry of Health, Labour and Welfare of Japan), a Grant-in-Aid for Scientific Research C (23K07409) from JSPS."

"Health and Labour Sciences Research Grants for Research on Measures for Intractable Diseases (from the Ministry of Health, Labour and Welfare of Japan), a Grant-in-Aid for Scientific Research C (23K07409) from JSPS"

6. Please provide a complete Data Availability Statement in the submission form, ensuring you include all necessary access information or a reason for why you are unable to make your data freely accessible. If your research concerns only data provided within your submission, please write "All data are in the manuscript and/or supporting information files" as your Data Availability Statement.

7. PLOS ONE now requires that authors provide the original uncropped and unadjusted images underlying all blot or gel results reported in a submission’s figures or Supporting Information files. This policy and the journal’s other requirements for blot/gel reporting and figure preparation are described in detail at https://journals.plos.org/plosone/s/figures#loc-blot-and-gel-reporting-requirements and https://journals.plos.org/plosone/s/figures#loc-preparing-figures-from-image-files. When you submit your revised manuscript, please ensure that your figures adhere fully to these guidelines and provide the original underlying images for all blot or gel data reported in your submission. See the following link for instructions on providing the original image data: https://journals.plos.org/plosone/s/figures#loc-original-images-for-blots-and-gels. 

8. Please amend your manuscript to include your abstract after the title page.

Reviewers' comments:

Reviewer's Responses to Questions

**Comments to the Author**

1. Is the manuscript technically sound, and do the data support the conclusions?

Reviewer #1: Yes

Reviewer #2: Yes

Reviewer #3: Yes

2. Has the statistical analysis been performed appropriately and rigorously? 

Reviewer #1: N/A

Reviewer #2: No

Reviewer #3: Yes

3. Have the authors made all data underlying the findings in their manuscript fully available?

Reviewer #1: Yes

Reviewer #2: Yes

Reviewer #3: Yes

4. Is the manuscript presented in an intelligible fashion and written in standard English?

Reviewer #1: Yes

Reviewer #2: Yes

Reviewer #3: Yes

5. Review Comments to the Author

Reviewer #1: The collection details of the liver sample have not been mentioned, such as the number of patients from whom the samples were obtained, the hospitals from which the samples were collected, how these samples were obtained from pateint, and other relevant details.

Please use more recent references, as out of the 34 references, 14 are older than 5 years.

Statistical analysis is not available; therefore, it is necessary for you to include statistical analysis in your research, especially when comparing samples and when you have a control group.

Additionally, no information has been provided regarding age and gender differences in the results.

Reviewer #2: GENERAL COMMENTS

The paper is commendably well-written, providing lucid explanations of both the disease itself and the objectives of the study, thereby ensuring accessibility for readers unacquainted with the subject matter. The authors adeptly contextualize their research within the existing literature, delineating the gaps they seek to address. The research's objective is articulated with clarity, underscoring its significance for enhancing understanding of the disease and hinting at its potential implications for designing novel therapeutic approaches. The selection and description of materials and methods are appropriate, facilitating replication. While the results are effectively presented and supplemented with figures and tables for enhanced comprehension, minor issues merit attention.

The findings convincingly demonstrate the presence of exogenous pathogen-associated antigens, particularly bacteria, in both tissue and serum samples obtained from PBC patients participating in the study. Importantly, the identified species align with previous reports, reinforcing the theory of environmental factors' involvement in the disease's pathogenesis. The ensuing discussion is well-informed, seamlessly integrating insights gleaned from the literature with the study's results. The authors conscientiously acknowledge the study's limitations, recognizing them as areas for improvement in future investigations.

Overall, the study successfully achieves its aim of identifying exogenous pathogens, presenting their results, and contributing to the understanding of the disease and its pathogenesis. Furthermore, it paves the way for further experimentation, underlining the imperative for continued research in this domain. While there are minor issues that warrant review and refinement in the final draft, the paper's substantive contributions to the field of PBC research warrant its publication, pending resolution of these issues, as it promises to catalyze advancements in future studies of PBC.

MINOR ISSUES AND RECOMMENDATIONS

*Materials and methods:

1. Remember always clarifying the significance of initials the first time they are mentioned, even if it sounds unnecessary (e.g., stained with HE, what does HE stand for?) (page 12, final line)

2. Regarding the 16S rRNA analysis (page 14, lines 1 to 3), is there any information on why those specific regions were selected or past studies that have used them? It would be good to give a little more context on why those regions are used for the analysis and references to past, similar studies that used them.

3. For the validation study (page 14, line 17), I would suggest adding a little bit on how the authors designed the primers. Software used and other details.

4. About statistics analysis. Did the authors use any kind of statistics? Even just descriptive statistics? If so, it should have been mentioned.

*Results:

1. About the table for the clinical characteristics of the patients (page 29), in the legend it says that the parameters are presented as the median for continuous variables, then I assume some statistics were used. As said before, all kinds of statistics should be mentioned.

2. Did the authors have patients fill in questionnaires? If so, it also should be mentioned.

3. DNA library in PBC sera paragraph (page 16), I think grammar can be improved in this section, for some reason I find it hard to understand some phrases (e.g., fig 1B shows that reads peak histogram in PBC samples?)

4. 16S rRNA metagenome analysis in sera (page 17, lines 7 to 9), there are two phrases with the same meaning in this section, it turns repetitive.

*Tables and figures:

1. This detail was something that I found mostly in tables and legends, but I will advise the authors to check the entire document for the style regarding the writing of species names, as in some parts of the document the species are written in italics, but in some they are not in italics. It may sound like a small thing but being consistent throughout the text is important.

2. Also in table 4, the writing for Sphingomonas panacis should be checked as the species (i.e., panacis) should always go all lower case.

3. Legend for figure 3 (Page 28, line 6), there is a phrase that sounds loose, like some information is missing or the phrase is incomplete.

4. In figure 2B, where is the before image for the hepatocyte area? It is mentioned in the legend that it is supposed to be a before and after image, but the figure only shows the after. Also, the scale bar is missing in this figure 2B.

5. In figure 3, do the samples have any kind of ID that could be put in the graphic to indicate to which sample each line corresponds? If so, it should be added to the image for better interpretation.

Reviewer #3: Following suggestions are recommended to further improve your manuscript;

1) kindly check out some recent plos one publications and format your manuscript as per journal guidelines.

2) kindly add line numbers.

3) kindly arrange all the intext and end bibliography as per journal guidelines.

4) it is recommended to arrange all the manuscript sections as per journal format.

5) kindly make a separate table for primer sequences instead of discussing them in paragraph.

6) it is suggested to correct the manuscript spacing as per journal format.

7) kindly arrange all the figures and figure legends as per journal guidelines, as they should be mentioned after the paragraph they are discussed in.

8) what does (mean 2578 reads) means throughout the manuscript.

9) kindly revise all the tables of manuscript to be in uniform format.

6. PLOS authors have the option to publish the peer review history of their article (what does this mean?). If published, this will include your full peer review and any attached files.

Reviewer #1: **Yes: **Souzan H Eassa

Reviewer #2: No

Reviewer #3: No

---

## [Author Response · Author response to Decision Letter 0]

24 Jul 2024

Responses to Reviewers:

We are grateful for the reviewers' very thoughtful and constructive suggestions. Most of their suggestions are reasonable and acceptable. Accordingly, we have additionally answered their queries. The followings are our point responses to the reviewers’ comments. 

Reviewer's Comments

Reviewer #1 (Comments to the Author):

 The collection details of the liver sample have not been mentioned, such as the number of patients from whom the samples were obtained, the hospitals from which the samples were collected, how these samples were obtained from pateint, and other relevant details.

Our answer:

We appreciate the important question. We have added the PBC sample collection sentence in Material and Methods session. The number of patients was mentioned in Result session.

Please use more recent references, as out of the 34 references, 14 are older than 5 years.

Our answer:

We thank for this reasonable comment. We have replaced them with more recent references. 

Statistical analysis is not available; therefore, it is necessary for you to include statistical analysis in your research, especially when comparing samples and when you have a control group.

Additionally, no information has been provided regarding age and gender differences in the results.

Our answer:

We appreciate the important question. We have created a new statistical analysis session and additionally described the methods used in this study.

Reviewer #2 (Comments to the Author):

The paper is commendably well-written, providing lucid explanations of both the disease itself and the objectives of the study, thereby ensuring accessibility for readers unacquainted with the subject matter. The authors adeptly contextualize their research within the existing literature, delineating the gaps they seek to address. The research's objective is articulated with clarity, underscoring its significance for enhancing understanding of the disease and hinting at its potential implications for designing novel therapeutic approaches. The selection and description of materials and methods are appropriate, facilitating replication. While the results are effectively presented and supplemented with figures and tables for enhanced comprehension, minor issues merit attention.

 The findings convincingly demonstrate the presence of exogenous pathogen-associated antigens, particularly bacteria, in both tissue and serum samples obtained from PBC patients participating in the study. Importantly, the identified species align with previous reports, reinforcing the theory of environmental factors' involvement in the disease's pathogenesis. The ensuing discussion is well-informed, seamlessly integrating insights gleaned from the literature with the study's results. The authors conscientiously acknowledge the study's limitations, recognizing them as areas for improvement in future investigations.

 Overall, the study successfully achieves its aim of identifying exogenous pathogens, presenting their results, and contributing to the understanding of the disease and its pathogenesis. Furthermore, it paves the way for further experimentation, underlining the imperative for continued research in this domain. While there are minor issues that warrant review and refinement in the final draft, the paper's substantive contributions to the field of PBC research warrant its publication, pending resolution of these issues, as it promises to catalyze advancements in future studies of PBC.

*Materials and methods:

1. Remember always clarifying the significance of initials the first time they are mentioned, even if it sounds unnecessary (e.g., stained with HE, what does HE stand for?) (page 12, final line)

Our answer:

We thank for this reasonable comment. We have corrected the wording appropriately, especially in the areas indicated.

2. Regarding the 16S rRNA analysis (page 14, lines 1 to 3), is there any information on why those specific regions were selected or past studies that have used them? It would be good to give a little more context on why those regions are used for the analysis and references to past, similar studies that used them.

Our answer:

We thank for this reasonable comment. We have added the information and reasons why certain areas were used, as you have indicated.

3. For the validation study (page 14, line 17), I would suggest adding a little bit on how the authors designed the primers. Software used and other details.

Our answer:

We thank for this reasonable comment. We have additionally described the primer design method and the software we used.

4. About statistics analysis. Did the authors use any kind of statistics? Even just descriptive statistics? If so, it should have been mentioned.

Our answer:

We appreciate the important question. We have created a new statistical analysis session and additionally described the methods used in this study.

*Results:

1. About the table for the clinical characteristics of the patients (page 29), in the legend it says that the parameters are presented as the median for continuous variables, then I assume some statistics were used. As said before, all kinds of statistics should be mentioned.

Our answer:

We appreciate the important question. We have created a new statistical analysis session and additionally described the methods.

2. Did the authors have patients fill in questionnaires? If so, it also should be mentioned.

Our answer:

We appreciate the important question. There are no questionnaires or other items listed for patients in this study.

3. DNA library in PBC sera paragraph (page 16), I think grammar can be improved in this section, for some reason I find it hard to understand some phrases (e.g., fig 1B shows that reads peak histogram in PBC samples?)

Our answer:

We thank for this reasonable comment. We have corrected the grammar in the areas pointed out.

4. 16S rRNA metagenome analysis in sera (page 17, lines 7 to 9), there are two phrases with the same meaning in this section, it turns repetitive.

Our answer:

We thank for this reasonable comment. We deleted the repeated part.

*Tables and figures:

1. This detail was something that I found mostly in tables and legends, but I will advise the authors to check the entire document for the style regarding the writing of species names, as in some parts of the document the species are written in italics, but in some they are not in italics. It may sound like a small thing but being consistent throughout the text is important.

2. Also in table 4, the writing for Sphingomonas panacis should be checked as the species (i.e., panacis) should always go all lower case.

Our answer:

We thank for this reasonable comment. We have corrected the notations noted.

3. Legend for figure 3 (Page 28, line 6), there is a phrase that sounds loose, like some information is missing or the phrase is incomplete.

Our answer:

We thank for this reasonable comment. We have added information such as sample names to Figure 3.

4. In figure 2B, where is the before image for the hepatocyte area? It is mentioned in the legend that it is supposed to be a before and after image, but the figure only shows the after. Also, the scale bar is missing in this figure 2B.

Our answer:

We thank for this reasonable comment. We have included pre- and post-LMD diagrams of the hepatocellular region. We have also added scale bar.

5. In figure 3, do the samples have any kind of ID that could be put in the graphic to indicate to which sample each line corresponds? If so, it should be added to the image for better interpretation.

Our answer:

We thank for this reasonable comment. We have added information such as sample names and ID to Figure 3.

Reviewer #3 (Comments to the Author):

Following suggestions are recommended to further improve your manuscript;

1) kindly check out some recent plos one publications and format your manuscript as per journal guidelines.

2) kindly add line numbers.

3) kindly arrange all the intext and end bibliography as per journal guidelines.

4) it is recommended to arrange all the manuscript sections as per journal format.

Our answer:

We thank for this reasonable comment. We revised our manuscript as per journal guidelines.

5) kindly make a separate table for primer sequences instead of discussing them in paragraph.

Our answer:

We thank for this reasonable comment. We made a separate table for primer sequences (S1 Table).

6) it is suggested to correct the manuscript spacing as per journal format.

7) kindly arrange all the figures and figure legends as per journal guidelines, as they should be mentioned after the paragraph they are discussed in.

Our answer:

We thank for this reasonable comment. We revised our manuscript as per journal guidelines.

8) what does (mean 2578 reads) means throughout the manuscript.

Our answer:

We appreciate the important question. These mean reads number refers to the number of short reads detected in RNA sequencing (16S rRNA metagenome). In other words, it shows how many specific regional reads were detected in a certain bacterium by sequencing.

9) kindly revise all the tables of manuscript to be in uniform format.

Our answer:

We thank for this reasonable comment. We revised our manuscript as per journal guidelines.

---

## [Decision Letter · Decision Letter 1]

2 Aug 2024

Identification of microbial antigens in liver tissues involved in the pathogenesis of primary biliary cholangitis using 16S rRNA metagenome analysis

PONE-D-24-05687R1

Dear Dr. Tomohiro Katsumi,

We’re pleased to inform you that your manuscript has been judged scientifically suitable for publication and will be formally accepted for publication once it meets all outstanding technical requirements.

Kind regards,

Hin Fung Tsang

Academic Editor

PLOS ONE

Additional Editor Comments (optional):

Reviewers' comments:

Reviewer's Responses to Questions

**Comments to the Author**

1. If the authors have adequately addressed your comments raised in a previous round of review and you feel that this manuscript is now acceptable for publication, you may indicate that here to bypass the “Comments to the Author” section, enter your conflict of interest statement in the “Confidential to Editor” section, and submit your "Accept" recommendation.

Reviewer #1: All comments have been addressed

Reviewer #2: All comments have been addressed

Reviewer #3: All comments have been addressed

2. Is the manuscript technically sound, and do the data support the conclusions?

Reviewer #1: Yes

Reviewer #2: (No Response)

Reviewer #3: Yes

3. Has the statistical analysis been performed appropriately and rigorously? 

Reviewer #1: Yes

Reviewer #2: (No Response)

Reviewer #3: Yes

4. Have the authors made all data underlying the findings in their manuscript fully available?

Reviewer #1: Yes

Reviewer #2: (No Response)

Reviewer #3: Yes

5. Is the manuscript presented in an intelligible fashion and written in standard English?

Reviewer #1: Yes

Reviewer #2: (No Response)

Reviewer #3: Yes

6. Review Comments to the Author

Reviewer #1: (No Response)

Reviewer #2: (No Response)

Reviewer #3: All the comments have been addressed. I would suggest to accept the manuscript. I must appreciate the author's findings and hard work.

7. PLOS authors have the option to publish the peer review history of their article (what does this mean?). If published, this will include your full peer review and any attached files.

Reviewer #1: **Yes: **Souzan H Eassa

Reviewer #2: No

Reviewer #3: No

---

## [Editor Report · Acceptance letter]

9 Aug 2024

PONE-D-24-05687R1 

PLOS ONE

Dear Dr. Katsumi, 

I'm pleased to inform you that your manuscript has been deemed suitable for publication in PLOS ONE. Congratulations! Your manuscript is now being handed over to our production team.

Kind regards, 

on behalf of

Dr. Hin Fung Tsang 

Academic Editor

PLOS ONE